# *BRCA1* Promoter Hypermethylation is Associated with Good Prognosis and Chemosensitivity in Triple-Negative Breast Cancer

**DOI:** 10.3390/cancers12040828

**Published:** 2020-03-30

**Authors:** William Jacot, Evelyne Lopez-Crapez, Caroline Mollevi, Florence Boissière-Michot, Joelle Simony-Lafontaine, Alexandre Ho-Pun-Cheung, Elodie Chartron, Charles Theillet, Antoinette Lemoine, Raphael Saffroy, Pierre-Jean Lamy, Séverine Guiu

**Affiliations:** 1Department of Medical Oncology, Montpellier Cancer Institute Val d’Aurelle, 208 rue des Apothicaires, F-34298 Montpellier, France; Elodie.Chartron@icm.unicancer.fr (E.C.); severine.guiu@icm.unicancer.fr (S.G.); 2Translational Research Unit, Montpellier Cancer Institute Val d’Aurelle, 208 rue des Apothicaires, F-34298 Montpellier, France; Evelyne.Crapez@icm.unicancer.fr (E.L.-C.); Florence.Boissiere@icm.unicancer.fr (F.B.-M.); Joelle.Simony@icm.unicancer.fr (J.S.-L.); Alexandre.Ho-pun-cheung@icm.unicancer.fr (A.H.-P.-C.); 3Faculty of Medicine, Montpellier University, 34090 Montpellier, France; 4Institut de Recherche en Cancérologie de Montpellier (IRCM), Inserm U1194, Université de Montpellier, Institut du Cancer Montpellier (ICM), F-34298 Montpellier, France; caroline.mollevi@icm.unicancer.fr (C.M.); charles.theillet@inserm.fr (C.T.); 5Biometrics Unit, Institut du Cancer Montpellier (ICM), Université de Montpellier, 208 rue des Apothicaires, F-34298 Montpellier, France; 6Department of Oncogenetics, APHP, GH Paris-Sud, Hôpital Paul Brousse, Inserm UMR-S 1193, Université Paris-Saclay, 14 Avenue Paul Vaillant Couturier, 94800 Villejuif, France; antoinette.lemoine@aphp.fr (A.L.); raphael.saffroy@aphp.fr (R.S.); 7Institut d’Analyse Génomique, Imagenome-Inovie, Clinique BeauSoleil, 34070 Montpellier, France; pierre-jean.lamy@labosud.fr; 8Biological Resources Center, Montpellier Cancer Institute Val d’Aurelle, F-34298 Montpellier, France

**Keywords:** triple-negative breast cancer, BRCA1, expression, promoter hypermethylation, prognosis, basal-like

## Abstract

The aberrant hypermethylation of *BRCA1* promoter CpG islands induces the decreased expression of BRCA1 (Breast Cancer 1) protein. It can be detected in sporadic breast cancer without *BRCA1* pathogenic variants, particularly in triple-negative breast cancers (TNBC). We investigated *BRCA1* hypermethylation status (by methylation-specific polymerase chain reaction (MS-PCR) and MassARRAY^®^ assays), and BRCA1 protein expression using immunohistochemistry (IHC), and their clinicopathological significance in 248 chemotherapy-naïve TNBC samples. Fifty-five tumors (22%) exhibited *BRCA1* promoter hypermethylation, with a high concordance rate between MS-PCR and MassARRAY^®^ results. Promoter hypermethylation was associated with reduced IHC BRCA1 protein expression (*p* = 0.005), and expression of Programmed death-ligand 1 protein (PD-L1) by tumor and immune cells (*p* = 0.03 and 0.011, respectively). A trend was found between promoter hypermethylation and basal marker staining (*p* = 0.058), and between BRCA1 expression and a basal-like phenotype. In multivariate analysis, relapse-free survival was significantly associated with N stage, adjuvant chemotherapy, and histological subtype. Overall survival was significantly associated with T and N stage, histology, and adjuvant chemotherapy. In addition, patients with tumors harboring *BRCA1* promoter hypermethylation derived the most benefit from adjuvant chemotherapy. In conclusion, *BRCA1* promoter hypermethylation is associated with TNBC sensitivity to adjuvant chemotherapy, basal-like features and PD-L1 expression. BRCA1 IHC expression is not a good surrogate marker for promoter hypermethylation and is not independently associated with prognosis. Association between promoter hypermethylation and sensitivity to Poly(ADP-ribose) polymerase PARP inhibitors needs to be evaluated in a specific series of patients.

## 1. Introduction

Triple-negative breast cancers (TNBCs) represent 15% of all breast cancers (BCs) and are defined by the absence of estrogen receptors (ERs), progesterone receptors (PRs), and Growth Factor Receptor-2 (HER2) overexpression/amplification [1,2]. Despite the good chemosensitivity of these tumors, patients with TNBC have a poor prognosis [1,3]. Based on hierarchical clustering, Perou et al. initially described five ‘intrinsic’ molecular BC subtypes, including a basal-like subtype [4,5]. Approximately 70% of basal-like BCs have a triple-negative phenotype (defined by immunohistochemistry [IHC]) and, using the PAM50 classifier, 80% of TNBCs can be classified as basal-like tumors [6]. An expanded surrogate immunopanel of markers (ER, PR, HER2, EGFR, and cytokeratin [CK] 5/6) provides a more specific IHC definition of basal-like BC [7]. Overall, 70% to 80% of BCs due to hereditary *BRCA1* pathogenic variants are TNBC [8,9,10]. ‘BRCAness’ is defined by the phenotypic similarities that some sporadic cancers share with those occurring in either *BRCA1*- or *BRCA2*-mutation carriers [11]. These tumors share a common deficiency in DNA repair.

Upregulation of their DNA repair capacity is a common mechanism used by cancer cells to survive DNA-damaging therapy [12]. Lack of efficient DNA repair by the simultaneous loss or inhibition of two DNA repair pathways causes synthetic lethality and cell death, thus representing an attractive approach for cancer therapy [13], emphasized by the clinical activity of poly(ADP-ribose) (PAR) polymerase 1 (PARP) inhibitors (PARP*i*) in patients with *BRCA*-mutated BC [14,15,16].

Identifying BRCA-deficient tumors is therefore of tremendous importance in this setting. The absence of BRCA1 nuclear expression correlates with high tumor grade and ER-negative tumors [17]. Absent or reduced BRCA1 expression in tumors without *BRCA1* pathogenic variants appears to be linked to hypermethylation of the *BRCA1* promoter region [18], a condition reported in 9.1–37% of sporadic BCs and associated with infiltrating ductal carcinoma type, high tumor grade (grade II-III), ER negativity, basal marker expression, younger age at diagnosis, and poor prognosis [18,19,20,21,22,23,24,25,26,27,28,29]. Thus, *BRCA1* promoter hypermethylation could be a marker of BRCA1 deficiency in the absence of *BRCA1* mutation, as these two events appear to be almost mutually exclusive [23,30,31,32,33,34], outside of the recently described association between a dominantly inherited 5’ UTR variant, classified as likely pathogenic, and *BRCA1* promoter hypermethylation [35]. In contrast, no *BRCA2* promoter methylation is implicated in breast carcinogenesis and rarely so in ovarian cancer [36]. Instead, BRCA2 expression has been proposed to be down-regulated by EMSY, a chromatin remodeling protein, shown to interact with the BRCA2 transactivation domain and to repress its transcription. The EMSY gene is amplified and overexpressed in 13% BC and 17% HGSOC and this is considered as a manifestation of BRCA2 inactivation in these tumors [37].

Outside of *BRCA1/2* pathogenic variants, there is to date no validated screening test to identify breast cancer patients who may derive the most benefit from PARP*i*. Recent data show that most non-*BRCA*-mutated TNBCs do not benefit from such drugs, while some non-TN *BRCA*-mutated tumors do respond to PARP*i* [38]. Moreover, two different groups [39,40] reported that BCs with epigenetically silenced *BRCA1* are sensitive to PARP*i* monotherapy, providing initial evidence to support the use of PARP*i* to treat selected sporadic *BRCA1*-inactivated BCs. *BRCA1* promoter hypermethylation is therefore a biomarker of interest in TNBC. In order to find a simple and reproducible test, previous studies evaluated whether BRCA1 protein expression assessed using IHC could act as a surrogate marker for *BRCA1* transcription. The correlation between levels of *BRCA1* transcription and Western blot protein quantification has been reported as weak [41]. In the study by Al-Mula et al., the same weak correlation was found between IHC protein evaluation and mRNA levels quantification, using real-time RT-PCR [42]. Reported correlations with epigenetic inactivation through promoter hypermethylation varied, from good correlations in high-grade ovarian serous carcinoma [43] to weak in BC [44]. BRCA1 IHC evaluation suffers from considerable run-to-run variability [41]. Recent conflicting data on the association between patient prognosis and BRCA1 IHC status or *BRCA1* promoter hypermethylation levels [27,43,45] mean a comprehensive evaluation of both as potential TNBC biomarkers is required. Identification of additional targets, such as immune checkpoint proteins, would allow better refinement of the definition and therapeutic targeting [46] of this tumor population. Indeed, an association has been described between genomic instability and sensitivity to immune checkpoint-targeted therapies [47]. Thus, PD-L1 expression in the context of BRCA1-deficient tumors could indicate a putative candidate population for PD-L1 inhibition [48].

Here, we simultaneously evaluated *BRCA1* methylation status and BRCA1 protein expression and their clinicopathological significance in a population of 248 sporadic TNBCs from patients without familial BC history or known germline *BRCA1* pathogenic variants, in order to evaluate the robustness of the IHC evaluation of protein expression as a surrogate endpoint of epigenetic inactivation, and to evaluate their association with prognosis and PD-L1 expression.

## 2. Results

### 2.1. Patient and Tumor Characteristics

Only cases with available BRCA1 IHC analysis and *BRCA1* promoter methylation status were selected for this study (*n* = 248; Figure 1).

Table 1 summarizes the main clinicopathological characteristics of this cohort, which were consistent with classical TNBC features. The patients’ median age was 57.8 years (range: 28.5–98.6 years). Ductal carcinoma was the most common histological type (82.9%), and 73.3% of patients received adjuvant chemotherapy (CT), while the remaining 26.7% of patients received adjuvant radiation therapy only, if clinically indicated. As per our guidelines, our patients received either anthracycline-based chemotherapy (four to six cycles of FEC100/EC 100/FAC65, 33.1% of the patients) or sequential anthracyclines and taxanes-based chemotherapy (three cycles of FEC100/EC100, followed by three cycles of three-weekly docetaxel or nine injections of weekly paclitaxel, 62.4% of the patients). Five patients received taxane-based chemotherapy, and one patient received three cycles of CMF followed by three cycles of taxanes, due to a previous history of anthracyclines treatment for a hematologic malignancy. Only one of our patients received an adjuvant platinum salts-based chemotherapy (six cycles of docetaxel—carboplatin, due to cardiovascular comorbidities precluding the use of anthracyclines). None of the patients received additional hormonal therapy, targeted therapy, or an investigational product.

### 2.2. In Situ BRCA1 IHC Analysis

In this study, we used the previously validated MS110 monoclonal antibody [43,49] to detect the expression of BRCA1 by IHC in a cohort of 349 TNBCs arrayed on six tissue microarrays (TMAs). The IHC procedure was previously optimized on full-face tissue sections, using various conditions of antigen retrieval, antibody dilution, amplification steps, and signal detection. All TMA sections were stained in the same run with the optimized procedure. Due to the heterogeneity of BRCA1 expression, samples scored as absent or equivocal were assessed subsequently on full-face sections. BRCA1 expression was considered uninterpretable for 31 cases, in which the tissue cores were lost during processing or showed nonspecific IHC staining. Twenty-nine of the 318 remaining cases (9.1%) were unclassifiable following Garg and Meisel criteria (for example, weak or moderate staining of 10% of tumor cell nuclei in the presence of a weak internal positive control). Of the remaining 289 cases, tumor DNA was not available for 41 cases. Overall, 248 cases were assessed for both BRCA1 protein expression and *BRCA1* promoter hypermethylation and constituted the analyzed data set (Figure 2). Following Garg and Meisel criteria, of 248 cases, 165 (66.5%), 70 (28.2%), and 13 (5.2%) displayed retained, absent, and equivocal BRCA1 staining, respectively (Appendix A).

### 2.3. Detection of BRCA1 Promoter Hypermethylation

*BRCA1* promoter hypermethylation status was evaluated by MS-PCR assays on the entire cohort. In addition, MassARRAY^®^ platform testing was carried out on 153 tumors for quantitative methylation analysis. MS-PCR revealed that 55/248 tumors (22.2%) exhibited *BRCA1* promoter hypermethylation. Among the 153 tumors analyzed by both MS-PCR and the EpiTyper^®^ MassARRAY^®^, concordance was high (*p* < 0.001), with only four discordant methylation statuses (Figure 2), when the threshold for global methylation percentage was set at 10% (global methylation percentages of 9.5, 6, 4, and 4).

### 2.4. BRCA1 Expression, BRCA1 Promoter Hypermethylation, and Clinicopathological Associations

Of the 248 TNBC cases assessed for both BRCA1 expression by IHC and *BRCA1* promoter hypermethylation using MS-PCR, 55 tumors were hypermethylated. Of these 55, 18 (32.7%), 9 (16.4%), and 28 (50.9%) had absent, equivocal, or retained BRCA1 staining, respectively. Of the 193 unmethylated cases, 52 (26.9%), 4 (2.1%), and 137 (71.0%) had absent, equivocal, or retained BRCA1 staining, respectively. A significant association was found between promoter hypermethylation assessed by MS-PCR and protein expression assessed by IHC, either classified into three categories (retained, equivocal, and absent, *p* < 0.001) or into two categories (retained vs. equivocal/lost, *p* = 0.005) (Table 2). A trend was found between promoter hypermethylation and a basal-like phenotype (CK5/6 and/or EGFR IHC staining, *p* = 0.058), and also between BRCA1 expression (BRCA1 absent/equivocal vs. retained) and a basal-like phenotype (*p* = 0.09).

The Venn diagram (Figure 3) summarizes the cross-distribution of the evaluated population across the three proposed biomarkers of BRCA1 deficiency, i.e., BRCA1 protein expression, *BRCA1* promoter hypermethylation, and basal-like phenotype. Among the 248 TNBC cases, three tumors with missing IHC basal marker statuses were excluded from the analysis and only 14.7% (36/245) of samples did not exhibit any of the three BRCA1 deficiency markers. There is an overlap of TNBC BRCA1-deficient specimens assessed by the three different measures; however, only 25 (10.2%) of the tumors displayed the 3 biomarkers of BRCA1 deficiency.

Regarding the immune checkpoint protein PD-L1, a significant association was found between *BRCA1* promoter hypermethylation and PD-L1 expression by tumor cells, using a 1% threshold (52.8% of hypermethylated tumors for tumors with < 1% cells expressing PD-L1 vs. 70% for tumors with ≥ 1% cells expressing PD-L1; *p* = 0.03), or by immune cells using a 50% threshold (34.7% of hypermethylated tumors in score three tumors vs. 17.9% in tumors with PD-L1 expression score 0–2; *p* = 0.011) (Table 2).

### 2.5. Survival Analyses

Using 24 October 2016 as the cut-off date, the median follow-up was 7.8 years (95% CI [7.3; 8.8]). Seventy-three deaths (five-year overall survival (OS): 80.6%, 95% CI [74.9; 85.1]) and 66 relapses occurred (five-year relapse-free survival (RFS): 73.0%, 95% CI [66.6;78.4]). The relapse pattern of our population was consistent with the previously reported temporal distribution of relapse risk [1,50], as most relapses occurred during the first three years of follow-up.

Results of the univariate analysis (regarding RFS and OS) in the overall population are summarized in Table 3. High pT and pN stages, a lobular subtype, and absence of adjuvant CT were significantly associated with shorter RFS. The five-year RFS rate was 79.9% and 71% (*p* = 0.112) in *BRCA1* promoter hypermethylated and non-hypermethylated groups, respectively. For OS, age, pT and pN stages, and the use of adjuvant CT were significant determinants.

In multivariate analysis in the overall population (Table 4A), shorter RFS was significantly associated with high pN stage, no adjuvant CT (5-year RFS: 81% with vs. 68% without adjuvant CT, *p* = 0.038), and histological subtype; OS was significantly associated with pT and pN stage, histology, and adjuvant CT.

To evaluate the impact of these clinicopathological parameters in patients receiving adjuvant CT, separate multivariate analyses were performed in the populations who received (Table 4B) and who did not receive adjuvant CT (Table 4C). 

In the adjuvant CT population (Table 4B), shorter RFS was significantly associated with high pN stage and *BRCA1* promoter hypermethylation (*p* = 0.021); OS was significantly associated with pT and pN stage, with a trend for *BRCA1* promoter hypermethylation (*p* = 0.052).

In the population without adjuvant CT (Table 4C), shorter RFS was significantly associated with high pN stage only; OS was significantly associated with pT, pN stage and histological subtype.

As BRCA1 protein expression was not considered in the multivariate analysis (p-value >0.15), we performed a second multivariate analysis integrating this variable, in order to test the robustness of the results (Appendix A). BRCA1 protein expression by IHC was not a statistically significant determinant of prognosis, whatever the population considered, and the previously identified variables remained significant.

Using the Kaplan–Myer survival curves (Figure 4), adjuvant CT benefit seemed to be restricted to the group of patients with BRCA1-deficient tumors; there was a significant difference between the prognosis of patients affected by an hypermethylated tumor (*p* = 0.024) under adjuvant CT, without difference in patients that did not received adjuvant CT (*p* = 0.570). A trend was seen for adjuvant CT-treated patients in case of BRCA1 negative/equivocal tumors by IHC (*p* = 0.092), without difference in patients that did not received adjuvant CT (*p* = 0.595). In addition, the independent prognostic information delivered by multivariate analysis of *BRCA1* promoter hypermethylation status, and not of IHC-assessed BRCA1 expression, which was also associated with a number of unclassified tumors using the Garg and Meisel criteria (17%), favors the consideration of *BRCA1* promoter hypermethylation status in this TNBC population (Table 3).

## 3. Discussion

Comprehensive and accurate determination of BRCA1 expression in TNBCs would allow for informed decisions and the biomarker-driven delivery of effective drugs such as platinum derivatives [51] or PARP*i* [15]. Here, we report a concomitant analysis of different methods of identifying BRCA1 deficiency outside *BRCA1* pathogenic variants, their association with clinicopathological characteristics, and their prognostic value in a large homogeneous series of non-metastatic, chemotherapy-naïve TNBCs with a long follow-up. We showed that BRCA1 protein expression, assessed by IHC, correlated with promoter hypermethylation status, and that both were associated with prognosis and CT efficacy. We also showed that IHC analysis was limited and was not a significant independent predictor of RFS in our series. *BRCA1* promoter hypermethylation appeared to be the most robust biomarker able to identify a particular subgroup of TNBCs without *BRCA1* pathogenic variants, that were associated with a better prognosis and sensitivity to adjuvant CT.

IHC is an efficient, valuable, and inexpensive technique widely used in pathology departments for diagnostic, prognostic, and theragnostic parameters. The identification of TNBCs with defective BRCA1 expression is a major step from a therapeutic point of view, as recently emphasized by the therapeutic impact of PARP*i* in *BRCA1/2*-mutated tumors [15,16]. However, to date, there is no definitive recommendation to carry out IHC evaluation of BRCA1 expression in order to robustly screen for pathogenic variants, or to identify additional populations of interest. Whereas some authors [43,49] reported reliable IHC modalities in high-grade serous ovarian carcinomas, others [41,52,53] argue that IHC staining is not robust enough to be used as a screening test for detection of BRCA1 dysfunction. In our hands, this methodology reported a 33.4% rate of negative/equivocal IHC staining, in accordance with previous data in the high-grade serous ovarian carcinoma population [43,49], but was unable to classify 9.1% of the tested samples. This failure rate is in accordance with technical limitations reported in the study by Milner et al. [41]

Although we found a significant association between the promoter hypermethylation assessed by the MS-PCR assay and protein expression assessed by IHC, we observed, alongside unclassifiable samples, both false-positive and false-negative cases. Thus, the discordances between the two approaches appear not to be linked to a lack of sensitivity of the antibody used in IHC, but rather to a lack of robustness of the IHC procedure itself, as underlined by Milner et al. [41] Indeed, the sensitivity and specificity of the MS110 monoclonal antibody has been well evaluated, and this antibody, despite its previous use in testing for associations between BRCA1 expression and *BRCA1* pathogenic variants (but not promoter hypermethylation), cannot be used as a single marker for the determination of BRCA1 expression levels. Based on these data, we believe that IHC with the MS110 monoclonal antibody cannot be used to detect the epigenetic inactivation of *BRCA1* in TNBC. 

The 5’ end of *BRCA1* is embedded in a large CpG-rich region and CpG sites close to the transcription start site that have been shown to contain strong promoter activity [54]. In addition, the hypermethylation of these sites has been associated with reduced BRCA1 protein and mRNA levels [55,56]. Evaluation of the promoter hypermethylation status could help identify more precisely a subgroup of BRCA1-deficient tumors that do not possess *BRCA1* pathogenic variants. In this study, we performed qualitative and quantitative methodology to assess the *BRCA1* methylation status in 153 tumor samples. Data were highly concordant, with only four conflicting tumor samples (i.e, hypermethylation detected by MS-PCR vs. no hypermethylation detected by the Epityper^®^ MassARRAY^®^ assay), using an arbitrary 10% methylation threshold for the Epityper^®^ MassARRAY^®^ assay. These discrepancies may be related to the different CpG sites analyzed. Indeed, MS-PCR assessed CpG sites close to the *BRCA1* transcription start site, whereas the MassARRAY^®^ approach selected mostly CpG sites in the intron 1 region. Moreover, all four samples identified as hypermethylated by MS-PCR, but as unmethylated by the MassARRAY^®^ platform, had a methylation percentage close to the fixed cut-off value of 10% for MassARRAY^®^ positivity. Nevertheless, as reported by Kondrashova et al. [57], the level of *BRCA1* promoter methylation appears to be important in the context of sensitivity to PARP*i*, and this quantitative method allows a rapid and low-cost analysis of a large number of CpG sites. However, in our series, as shown in Figure 2, tumors with high percentages of methylation were rare (12 (7.84%), 9 (5.88%) and 2 (1.31%) tumors if considering 30%, 35%, and 50% cut-offs, respectively). Such low frequency precludes all robust statistical evaluation in our series, and warrants additional evaluation in large cohorts, and in the context of PARP*i* treatment of breast/ovarian carcinoma.

In our population, assessment of *BRCA1* promoter hypermethylation status identified a population deriving the most benefit from adjuvant CT. The association we identified between *BRCA1* promoter hypermethylation and prognosis after adjuvant CT is in accordance with previously published preclinical and clinical data [58,59]. We observed that the patients who benefit most from adjuvant CT are those with BRCA1-defective tumors, validating the results published in 2013 by Xu et al. [60] These results differ from those published by Tutt et al. in the TNT trial [61]. In this randomized phase II trial, *BRCA1* promoter hypermethylation was not predictive of an additional benefit of carboplatin over docetaxel in first-line treatment of TNBC. Indeed, BRCA1-defective tumors are known to be more sensitive to platinum salts; however, in this study, the methylation status of the tumors was assessed on archived primary tumors, sampled before patients had received adjuvant CT, and the trial addressed the issue of sensitivity of metastatic disease to first-line CT. A change in the methylation status of the promoter under selection pressure could explain this absence of association. Conversely, in our series, methylation status analysis was performed on the primary tumor, removed some weeks before the initiation of adjuvant CT, and none of our patients had metastatic or pretreated disease. Our results suggest that identification of BRCA1-defective tumors could allow the delivery of adjuvant CT to the sensitive population and avoid its use in the rest of the population, where the prognosis did not differ between patients who received or did not receive adjuvant CT. In our series, only one patient received an adjuvant platinum salts-based chemotherapy. In addition, the tumor was not hypermethylated, and expressed BRCA1 on the IHC evaluation. Thus, our result cannot be the result of an imbalance in platinum salts use between the two groups. Another point to consider is the absence of adjuvant CT in 26.7% of our population, mainly due to age, histological grade and/or the patient’s decision. While these differences between CT-treated and untreated patients could induce a bias in the survival analyses, these specific variables have been tested in the multivariate analysis, and were not retained as independent determinants of prognosis, strengthening our conclusions. Considering these results, reappraisal of the predictive value of *BRCA1* promoter methylation status prior to the initiation of CT for metastatic disease appears necessary.

Identifying all defective tumors remains the major challenge. *BRCA1* promoter hypermethylation does not represent the only mechanism of BRCA1 deficiency outside pathogenic variants, and patients with tumors with a loss of BRCA1 protein expression derived benefit from adjuvant CT in our series. However, IHC screening failed to identify *BRCA1* hypermethylated tumors. It is thus critical to identify other mechanisms responsible for the loss of BRCA1 expression in our population. Id4, a member of the Id (inhibitor of DNA binding) family of proteins, has been demonstrated to downregulate the expression of BRCA1 in in vitro and in-patient tumor samples [62,63,64,65,66]. Other epigenetic regulations, such as microRNAs (miRNAs), could also explain these results. miRNAs are small non-coding RNAs that bind to the 3’ untranslated (3’UTR) region of target messenger RNAs (mRNAs); they are known to regulate gene expression and are frequently deregulated in BC. Different miRNAs, such as miR-146, 218, 335, 342, 498, 548, or 638, regulate BRCA1 mRNA expression, and could contribute to the population of IHC BRCA1-negative tumors [63,67,68,69,70]. A critical comprehensive appraisal of these additional factors influencing BRCA1 expression is necessary, in order to better define the tumor population sensitive to CT. Additional efforts are necessary to determine the potential impact of additional events, such as loss of heterozygosity, on the population exhibiting *BRCA1* promoter hypermethylation [32].

In the absence of recurrent druggable targets in TNBCs, a considerable effort is presently focused on targeting immune checkpoints in this disease, with significant results [46]. One of the key determinants of immunotherapy activity appears to be linked to the tumor mutational burden and genome instability [47]. We report, to our knowledge for the first time, a significant association between *BRCA1* promoter hypermethylation and PD-L1 expression, in both tumor and immune cells. In the study by Sobral-Leite et al., PD-L1 expression showed a modest but significant positive association with the number of silent mutations in TNBC [71]. However, PD-L1 was not associated with BRCA1-like status evaluated using multiplex ligation-dependent probe amplification, an assay unable to identify methylation status. Zhu et al. also did not observe a significant association between *BRCA1* promoter hypermethylation and PD-L1 expression, in a series of 112 ovarian carcinomas [72]. Our findings, if validated in an independent series, could lead to the use of a combination of treatments targeting synthetic lethality in TNBC, for example by combining a DNA-damaging agent and/or a PARPi to sensitize tumor cells to immune checkpoint inhibitors; such combinations are currently undergoing testing in patients with *BRCA*-mutated tumors. However, a mandatory first step would be a reappraisal of the predictive value of *BRCA1* promoter hypermethylation in the context of treatment by PARP inhibitors.

## 4. Materials and Methods 

### 4.1. Objectives

The primary endpoint of the study was the evaluation of the robustness of BRCA1 protein expression assessed by IHC to predict *BRCA1* promoter methylation status, assessed by the methylation-specific polymerase chain reaction (MS-PCR) assay. The key secondary endpoint was the evaluation of the concordance between the determination of *BRCA1* promoter methylation status by the quantitative EpiTyper^®^ MassARRAY^®^ (Agena Bioscience, Sans Diego, CA, USA) and by MS-PCR. Other secondary endpoints were the evaluation of the association between *BRCA1* promoter methylation status assessed by MS-PCR and clinicopathological variables, and the evaluation of the impact of these variables on overall survival (OS) and relapse-free survival (RFS).

### 4.2. Patients and Tumor Samples

A total of 1695 consecutive patients with BC, who were referred to our institution between 2002 and 2010, were prospectively entered into the database of a dedicated tumor biobank (Biobank number BB-0033-00059). Samples were isolated from frozen, histologically proven, and macro-dissected invasive BC specimens, that were primarily handled for ER and PR testing using the dextran-coated charcoal (DCC) method, as previously described [73,74], or for uPA/PAI-1 quantification with the Femtelle^®^ test. Tumors were considered ER and PR negative when the receptor concentrations were lower than 10 fmol/mg of protein (using the DCC assay), or when < 10% tumor cells were positive for IHC staining [75]. HER2 status was determined based on HER2 protein expression levels evaluated by IHC, using the A485 monoclonal antibody (Dako, Denmark). Tumors with HER2 scores of 0 and 1+ were considered HER2 negative. In tumors with equivocal HER2 IHC test results (2+), gene amplification was evaluated using fluorescence or chromogenic in situ hybridization. Specimens with HER2 3+ scores were considered HER2 positive. Finally, 248 primary non-metastatic TNBCs, without neoadjuvant treatment, were selected for this study, based on the successful determination of their EGFR, CK5/6, and BRCA1 IHC status, the absence of a known *BRCA1* pathogenic variant (patients either tested and without pathogenic variant, or without oncogenetic testing), as well as their *BRCA1* promoter methylation status. Each individual treatment proposal was in accordance with our institution guidelines [76]. The clinicopathological characteristics and treatment of the 248 patients included in this study are summarized in Table 1. This study was reviewed and approved by the Montpellier Cancer Institute Institutional Review Board (ID number ICM-CORT-2015-11). All patients gave their written, informed consent. As part of the study evaluated the prognostic impact of biological markers, this manuscript adheres to the REMARK guidelines.

### 4.3. Tissue Microarray and Immunohistochemistry

Tissue blocks appearing to have enough material upon gross inspection were selected from the Biological Resource Center of the Montpellier Cancer Institute (Biobank number BB-0033-00059). Hematoxylin–eosin-safranin (HES)-stained sections were evaluated by a pathologist for the presence of carcinoma. Two representative tumor areas, to be used for tissue microarray (TMA) construction, were identified on each slide. The tissues corresponding to selected areas were sampled using a manual arraying instrument (Manual Tissue Arrayer 1, Beecher Instruments, Sun Prairie, WI, USA). The sampling consisted of two malignant cores (1 mm in diameter) from different areas of the tumor, placed at specified coordinates. When possible, normal breast epithelium was also selected as an internal control. Finally, 349 tumors were sampled in six TMA blocks. After completion of the arraying process, TMA blocks were sectioned at a thickness of 4 μm. One section was stained with HES and the others were used for IHC. TMA sections to be stained for BRCA1, CK5/6, or PD-L1 underwent PTLink pre-treatment (Dako/Agilent, Santa Clara, CA, USA), allowing simultaneous de-paraffinization and antigen retrieval. Heat-induced antigen retrieval was executed for 15 min in High pH Buffer (Dako/Agilent, Santa Clara, CA, USA) at 95 °C. TMA sections to be stained for EGFR were deparaffinized and rehydrated prior to antigen retrieval with proteinase K. Following antigen retrieval, endogenous peroxidase was quenched using Flex Peroxidase Block (Dako/Agilent, Santa Clara, CA, USA), for 5 min at room temperature. Slides were then incubated at room temperature with antibodies raised against human BRCA1 (mouse monoclonal, clone MS110, 1:100, 20 min, Calbiochem, San Diego, CA, USA), human CK 5/6 (mouse monoclonal, clone 6D5/16 B4, 1:100, 20 min, Dako/Agilent, Santa Clara, CA, USA), human PD-L1 (rabbit monoclonal, clone SP142, 1:200, 30 min, Roche, Penzberg, Germany), or human EGFR (mouse monoclonal, clone 31G7, 1:50, 20 min, Invitrogen, Carlsbad, CA, United Stades). An amplification step was used for BRCA1 and PD-L1 immunostaining (mouse or rabbit linker, respectively, Dako/Agilent, Santa Clara, CA, USA). After two rinses in buffer, the slides were incubated with a horseradish peroxidase-labeled polymer coupled to secondary anti-mouse and anti-rabbit antibodies for 20 min, followed by application of 3,3′-diaminobenzidine for 10 min as a substrate. Counterstaining was performed using Flex Hematoxylin (Dako/Agilent, Santa Clara, CA, USA), after which slides were washed under tap water for 5 min. Finally, slides were mounted with a coverslip after dehydration.

### 4.4. TMA Analysis/Interpretation

TMA sections were analyzed independently by two trained observers, both blinded to the clinicopathological characteristics and patient outcomes at the time of scoring. For BRCA1 staining, the percentages of stained nuclei at each intensity (weak, moderate, and strong) were collected and used to score each tumor as BRCA1 loss, BRCA1 equivocal, or BRCA1 retained, following the Garg and Meisel criteria [43,49]. In the case of disagreement, the BRCA status was determined by consensus after simultaneous dual reexamination. Stromal cells served as internal positive controls and validated the staining. If BRCA1 expression was scored as absent or equivocal, the BRCA1 IHC procedure was carried out on corresponding full-face tissue sections, followed by the same scoring procedure, and scores obtained in the whole section were used. CK 5/6 and EGFR expression was scored according to the percentage of positive tumor cells, irrespective of the staining intensity. Tumors were determined as basal-like if ≥1% of tumor cells expressed CK5/6 and/or EGFR. PD-L1 expression was evaluated independently in invasive cancer cells (percentage of tumor cells exhibiting membranous staining, regardless of its intensity) and the immune cell component (percentage of PD-L1-positive cells, regardless of its cellular localization or intensity; specimens were scored as 0, 1, 2, or 3 if 0%, <10%, 10–50%, or ≥50% of immune cells were PD-L1-positive, respectively).

### 4.5. Tissue Processing and DNA Extraction 

DNA was extracted from frozen tumor tissue samples, as previously described [25,73], with the QIAamp DNA Mini Kit (Qiagen GmbH, Hilden, Germany), according to the manufacturer’s instructions.

### 4.6. BRCA1 Promoter Methylation Status

DNA methylation patterns at the CpG islands of the *BRCA1* promoter were assessed, using the MS-PCR assay previously described [25,32,77]. This method distinguishes between unmethylated and hypermethylated alleles on the basis of sequence changes following bisulfite treatment of DNA, that converts only unmethylated cytosines to uracil. Bisulfite treatment was performed using the EpiTect Bisulfite Kit (Qiagen GmbH, Hilden, Germany). PCRs were performed on an Eppendorf Mastercycler^®^ (Eppendorf, Hamburg, Germany), with the EpiTect MSP-PCR Kit (QIAGEN GmbH, Hilden, Germany), and specific primers designed for hypermethylated or unmethylated *BRCA1* DNA sequences [32]. This MS-PCR analyzed a total of seven CpG sites, located at −37, −29, −21, −19, +16, +19, and +27, relative to the *BRCA1* exon 1A transcription start site (Figure 1). The primers for the methylated reaction generated a 75-bp-long amplicon and the primers for the unmethylated amplified an 86-bp-long product. The EpiTect PCR Control DNA Set (Qiagen Hindel, Germany), containing both bisulfite-converted methylated and unmethylated DNA and unconverted unmethylated DNA, were also added as MS-PCR controls. Seven µL of each PCR product was loaded directly onto an 1% agarose + 3% NuSieve GTG agarose gel, stained with 1 µL/10 mL SYBR^®^ Safe DNA gel stain, and visualized under UV light. Samples were scored as hypermethylated only when a distinct band was present on the gel. If DNA samples presented low-intensity bands in the agarose gel, PCR with 6-FAM™-labeled or HEX™-labeled forward primers, amplifying specifically hypermethylated or unmethylated sequences, respectively, was carried out. DNA fragment analysis was performed by capillary electrophoresis on an Applied Biosystems 3130 Genetic Analyzer. Analyses were conducted with the GeneMapper Software v3.7 (Apllied Biosystems, Foster City, CA, USA).

The biochemistry of the EpiTyper^®^ MassARRAY^®^ starts with the bisulfite treatment of genomic DNA, followed by PCR amplification of target regions. The reverse primers contain a T7 promoter tag. Next, in vitro RNA transcription is performed, followed by base-specific RNA cleavage. Finally, the cleavage products are analyzed using MALDI-TOF mass spectrometry (MassARRAY Analyzer). The methylated and unmethylated cytosine residues in the original genomic DNA are easily distinguished using EpiTYPER Software. Three amplicons were analyzed to cover 13 CpGs sites, located at −166, −134, −127, +177, +195, +222, +235, +256, +292, +294, +399, +402, and +457, relative to the *BRCA1* exon 1A transcription start site (Figure 5).

### 4.7. Statistical Analysis

Categorical variables were presented as frequency distributions, and continuous variables as medians and ranges. Categorical variables were compared with the Pearson’s chi-square or Fisher’s exact test. OS was defined as the time between the date of surgery and the date of death (whatever the cause). Patients lost to follow-up were censored at the last documented visit. RFS was defined as the time between the date of surgery and the date of recurrence. Patients alive at the last follow-up without recurrence and patients lost to follow-up were censored at the last documented visit. Patients who died without recurrence were censored at the date of death. The Kaplan–Meier method was used to estimate the OS and RFS rates. Differences in survival rates were compared using the log–rank test. Multivariate analyses were performed using the Cox proportional hazard model, using variables with a *p*-value < 0.15 in univariate analysis. Hazard ratios (HR) are given with their 95% confidence interval (95% CI). Statistical analyses were performed with STATA 13.0 (StatCorp, College Station, TX, USA).

## 5. Conclusions

*BRCA1* promoter hypermethylation is associated with basal-like features and PD-L1 expression, and is independently associated with a better prognosis in TNBC, especially in the subgroup of patients treated with adjuvant CT. IHC evaluation of BRCA1 expression appears to be difficult to standardize and implement, considering the impact of the pre-analytical and analytical variables, even though the detected protein expression appears to be associated with the *BRCA1* hypermethylation status of the tumors. In our series, it did not appear to be independently associated with prognosis. The association between promoter hypermethylation and sensitivity to PARP inhibitors needs to be evaluated in a specific series of patients.

## Figures and Tables

**Figure 1 cancers-12-00828-f001:**
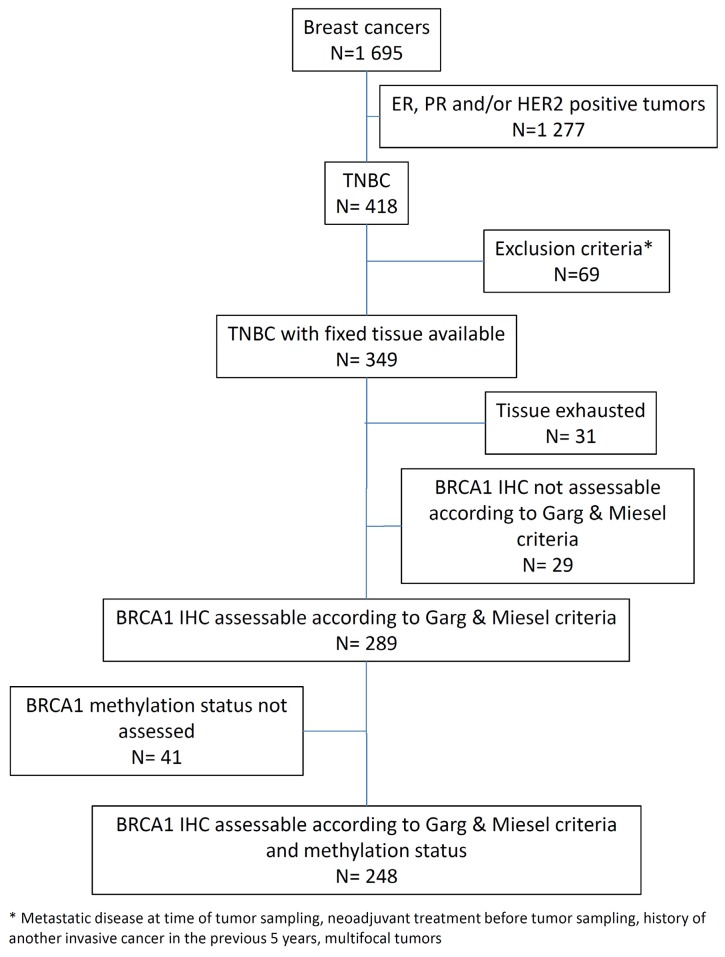
Consort diagram.

**Figure 2 cancers-12-00828-f002:**
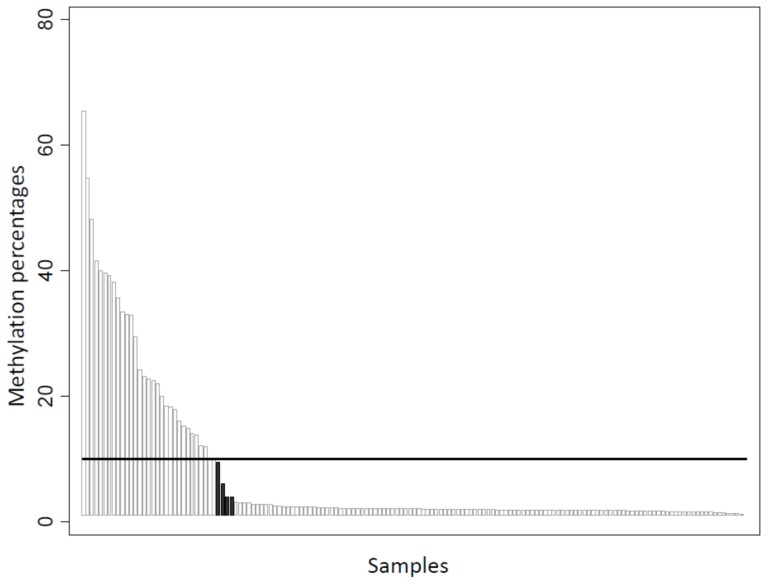
Correlation between methylation-specific PCR and EpiTYPER® MassARRAY® methylation evaluation (*n* = 153). Black columns indicate discordant cases (*n* = 4, methylation percentage assessed by MassARRAY® = 9.5, 6, 4, and 4), using a positivity cut-off of 10%.

**Figure 3 cancers-12-00828-f003:**
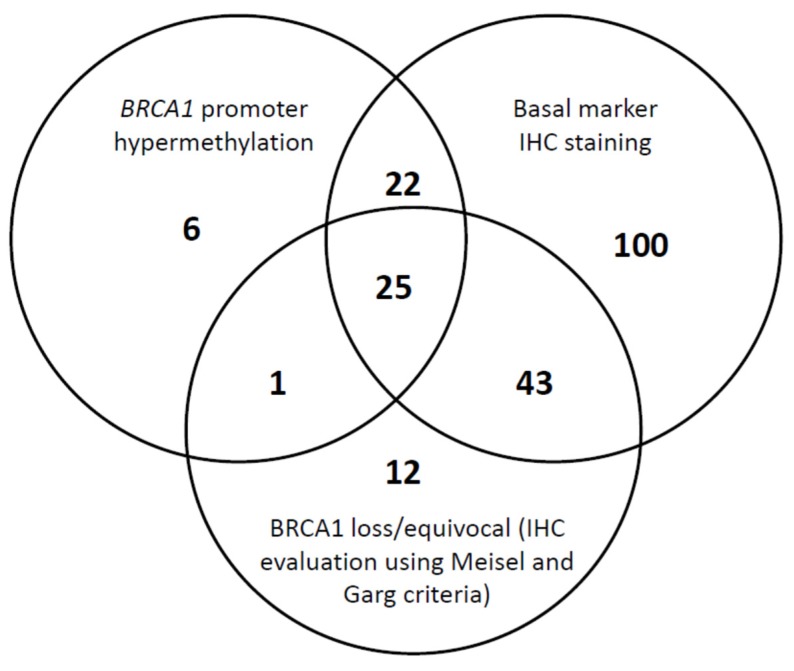
Shared BRCA1 (Breast Cancer 1) deficiency measures in triple negative breast cancers.

**Figure 4 cancers-12-00828-f004:**
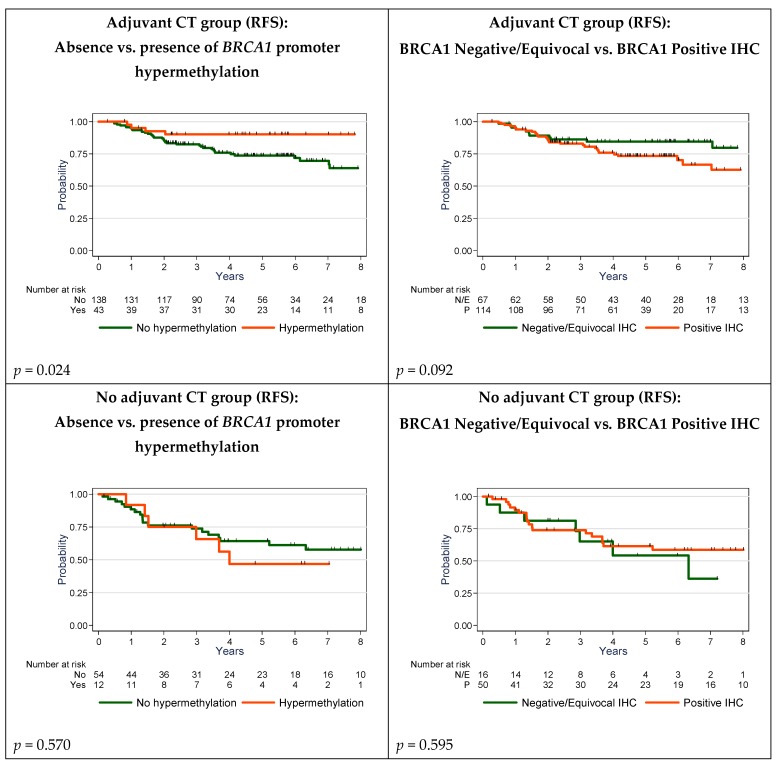
Relapse-Free Survival (RFS) in adjuvant chemotherapy-treated and untreated patients by BRCA1 immunohistochemistry (IHC) expression and *BRCA1* promoter hypermethylation.

**Figure 5 cancers-12-00828-f005:**
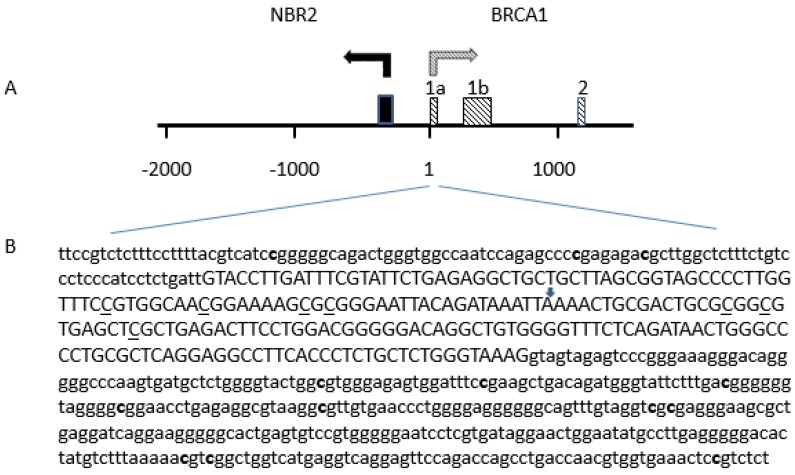
Distribution of CpG sites analyzed in the promoter region of the *BRCA1* gene. (**A**) The BRCA1–NBR2 locus: the position of the first exons of the BRCA1 is indicated by shaded boxes. The first exon of NBR2 gene is indicated by a black box. (**B**) Position of CpG sites analyzed Cytosine on CpG sites tested by MS-PCR are underlined. Cytosine on CpG sites tested by MassARRAY® EpiTYPER® assay are in bold. The positive strand of the BRCA1 gene is shown based on GenBank accession number U37574. The transcription start site of the BRCA1 exon 1A is marked by an arrow. Exon 1A sequence is in capital, and both 5’upstream sequence and intron 1 sequence are in lower case characters.

**Table 1 cancers-12-00828-t001:** Patients and tumors characteristics.

		*n* = 248	%
**Age (years)**			
	Median (range)	57.8	28.5–98.6
**Tumor Size**			
	T1	111	44.8
	T2	117	47.2
	T3/T4	20	8.0
**Nodal Status**			
	N-	162	65.3
	N+	86	34.7
**Tumor Grade**			
	1–2	59	24.2
	3	185	75.8
	Missing	4	
**Histology**			
	Ductal	204	82.9
	Lobular	12	4.9
	Others	30	12.2
	Missing	2	
**Basal-like Phetotype**			
	Nul	55	22.5
	Basal-like	190	77.5
	Missing	3	
**BRCA1 IHC Expression**			
	BRCA1-/	70	28.2
	BRCA1+	165	66.5
	Equivocal	13	5.2
***BRCA1* Promoter Hypermethylation**			
	No	193	77.8
	Yes	55	22.2
**PD-L1 Expression (tumor cells)**			
	< 1%	100	43.5
	≥ 1%	130	56.5
	Missing	18	
**PD-L1 Expression (Immune cells)**			
	≤ 50%	179	78.5
	> 50%	49	21.5
	Missing	20	
**Adjuvant Chemotherapy**			
	No	66	26.7
	Yes	181	73.3
	Missing	1	
**Relapse**			
	No	182	73.4
	Yes	66	26.6
**Status**			
	Alive	175	70.6
	Dead	73	29.4

BRCA1: Breast Cancer 1; EGFR: Epithelial Growth Factor Receptor. Basal-like phenotype was considered in case of a positive staining for either cytokeratins 5/6 and/or EGFR (≥1% tumor cells stained in IHC), while nul phenotype was considered in the absence of cytokeratins 5/6 and EGFR staining.

**Table 2 cancers-12-00828-t002:** Univariate clinicopathological correlations with *BRCA1* promoter methylation using MS-PCR.

	*BRCA1* Promoter Methylation Status				
	No		Yes		*p*
	*n* = 193	%	*n* = 55	%	
***BRCA1* Methylation (MassARRAY, *n* = 153)**					<0.001
Hypermethylated	0	0	31	88.6	
Unmethylated	118	100	4	11.4	
**BRCA1 IHC Expression**					<0.001
BRCA1-	52	26.9	18	32.7	
BRCA1+	137	71.0	28	50.9	
Equivocal	4	2.1	9	16.4	
					0.005
BRCA1-/ Equivocal	56	29.0	27	49.1	
BRCA1+	137	71.0	28	50.9	
**Basal-Like Phetotype**					0.058
Nul	48	25.1	7	13.0	
Basal-like	143	74.9	47	87.0	
**PD-L1 Expression (Tumor Cells)**					0.030
<1%	85	47.2	15	30.0	
≥1%	95	52.8	35	70.0	
**PD-L1 Expression (Immune Cells)**					0.011
0/1/2	147	82.1	32	65.3	
3	32	17.9	17	34.7	

**Table 3 cancers-12-00828-t003:** Univariate analysis.

(*n* = 248)		Relapse-Free Survival (RFS)			Oveall Survival (OS)		
	N	Events	5-Year RFS	Hazard Ratio [95% CI]	Events	5-Year OS	Hazard Ratio [95% CI]
**Age (Years)**							
<55	108	23	79.6	1	22	87.4	1
≥55	140	43	68.0	1.52 [0.91; 2.52]	51	75.4	2.00 [1.21; 3.30]
		*p* = 0.103			*p* = 0.005		
**Tumor Size**							
pT1	111	16	85.3	1	17	90.6	1
pT2	117	39	66.9	2.46 [1.38; 4.41]	43	75.7	2.60 [1.48; 4.56]
pT3/pT4	20	11	44.9	4.88 [2.26; 10.5]	13	54.5	6.02 [2.91; 12.4]
		*p* < 0.001			*p* < 0.001		
**Nodal Status**							
pN-	162	24	85.8	1	35	88.9	1
pN+	86	42	50.9	3.94 [2.39; 6.52]	38	65.2	2.45 [1.55; 3.88]
		*p* < 0.001			*p* < 0.001		
**Histological Grade (SBR)**							
1–2	59	17	76.5	1	20	84.3	1
3	185	49	71.4	1.05 [0.60; 1.82]	52	78.9	1.05 [0.62; 1.76]
		*p* = 0.875			*p* = 0.863		
**Histology**							
Ductal	204	56	71.9	1	65	77.8	1
Lobular	12	6	54.7	1.94 [0.83; 4.50]	4	83.3	0.99 [0.36; 2.73]
Other	30	3	89.2	0.30 [0.09; 0.97]	4	96.7	0.33 [0.12; 0.92]
		*p* = 0.021			*p* = 0.083		
**Adjuvant Chemotherapy**							
No	66	25	60.5	1	36	62.9	1
Yes	181	41	77.7	0.53 [0.32; 0.87]	37	86.9	0.34 [0.22; 0.54]
		*p* = 0.011			*p* < 0.001		
**Basal-Like Phenotype**							
Nul	55	16	71.9	1	18	78.8	1
Basal-like	190	48	73.5	0.89 [0.50; 1.57]	54	81.2	0.91 [0.53; 1.55]
		*p* = 0.690			*p* = 0.723		
**BRCA1 IHC Expression**							
BRCA1−/Equivocal	83	18	79.2	1	21	82.8	1
BRCA1+	165	48	69.7	1.47 [0.85; 2.53]	52	79.4	1.39 [0.84; 2.31]
		*p* = 0.162			*p* = 0.199		
***BRCA1* Promoter Hypermethylation**							
No	193	56	71.0	1	60	79.4	1
Yes	55	10	79.9	0.58 [0.30; 1.14]	13	84.5	0.74 [0.40; 1.34]
		*p* = 0.112			*p* = 0.318		
**PD-L1 expression (tumor cells)**							
<1%	100	32	69.4	1	34	81.4	1
≥1%	130	30	74.4	0.74 [0.45; 1.22]	35	79.5	0.89 [0.56; 1.44]
		*p* = 0.242			*p* = 0.640		
**PD-L1 expression (Immune cells)**							
0/1/2	179	48	73.4	1	57	79.8	1
3	49	12	71.4	0.88 [0.47; 1.66]	10	85.9	0.66 [0.34; 1.30]
		*p* = 0.703			*p* = 0.227		

**Table 4 cancers-12-00828-t004:** Multivariate analysis (Relapse-Free Sruvival).

**A. Multivariate analysis: Overall population (n = 248)**
	Relapse-Free Survival	Overall Survival
	Hazard Ratio [95% CI]	*p*-Value	Hazard Ratio [95% CI]	*p*-Value
**Tumor Size**				0.003
T1			1	
T2			2.00 [1.12; 3.59]	
T3/T4			4.21 [1.81; 9.81]	
**Nodal Status**		<0.001		0.003
N-	1		1	
N+	4.99 [2.94; 8.49]		2.30 [1.35; 3.92]	
**Histology**		0.043		0.005
Ductal	1		1	
Lobular	1.11 [0.47; 2.62]		0.45 [0.15; 1.32]	
Other	0.30 [0.09; 0.95]		0.27 [0.09; 0.76]	
**Adjuvant Chemotherapy**		<0.001		<0.001
No	1		1	
Yes	0.39 [0.23; 0.65]		0.30 [0.18; 0.48]	
**B. Multivariate Analysis: Adjuvant Chemotherapy = Yes**
**(*n* = 181)**	**Relapse-Free Survival**	**Overall Survival**
	**Hazard Ratio [95% CI]**	****p**-Value**	**Hazard Ratio [95% CI]**	***p*-Value**
**Tumor size**				0.033
T1			1	
T2			2.74 [1.22; 6.16]	
T3/T4			2.32 [0.65; 8.24]	
**Nodal Status**		<0.001		0.032
N−	1		1	
N+	3.50 [1.84; 6.69]		2.12 [1.06; 4.23]	
***BRCA1* Promoter Hypermethylation**		0.021		0.052
No	1		1	
Yes	0.35 [0.12; 0.98]		0.39 [0.14; 1.13]	
**C. Multivariate Analysis: Adjuvant Chemotherapy = No**
**(*n* = 66)**	**Relapse-Free Survival**	**Overall Survival**
	**Hazard Ratio [95% CI]**	***p*-Value**	**Hazard Ratio [95% CI]**	***p*-Value**
**Tumor Size**				0.017
T1			1	
T2			1.35 [0.58; 3.14]	
T3/T4			5.18 [1.61; 16.7]	
**Nodal Status**		<0.001		0.019
N−	1		1	
N+	6.81 [2.99; 15.5]		2.84 [1.20; 6.70]	
**Histology**				0.020
Ductal			1	
Lobular			0.31 [0.07; 1.39]	
Other			0.22 [0.05; 0.96]

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
