# Peer review of "BRCA1 Promoter Hypermethylation is Associated with Good Prognosis and Chemosensitivity in Triple-Negative Breast Cancer"

_cancers, 2020, doi:10.3390/cancers12040828_

Round 1
Reviewer 1 Report
Here the authors compile and compare the clinicopathological associaitions of TNBCs classified base on either BRCA1 methylation or BRCA1 IHC protein expression status.
The title should be edited in order to be consistent with the results reported in lines 146-148 (no significant association of basal-like phenotype with either BRCA1 IHC or methylation status) , 189-192 (significant difference between the prognosis of patients with and those without BRCA1-deficient tumors (assessed either by BRCA1 promoter methylation status or by protein expression) subjected to adjuvant CT.
lines: 122-125: Inclusion of representative images for (retained, absent and equivocal) BRCA1 staining would be helpful to understand the scoring system used by the authors
lines 128-129: What selection criteria did the authors follow to choose those 153 out of 248 tumours for quantitative methylation analysis?
lines 392-393: The authors clearly state that 'samples were scored as methylated only when a distinct band was present on the gel'. However it is not clearly stated when samples were scored as hypermethylated.
Minor comments
lines 23-24: The sentence 'Aberrant hypermethylation of BRCA1 promoter CpG islands, decreasing expression of BRCA1' can be detected...is not very clear. Can it be rephrased, please?
lines 347-351: Do these incubations take place at room temperature?
Author Response
Here the authors compile and compare the clinicopathological associaitions of TNBCs classified base on either BRCA1 methylation or BRCA1 IHC protein expression status.
The title should be edited in order to be consistent with the results reported in lines 146-148 (no significant association of basal-like phenotype with either BRCA1 IHC or methylation status) , 189-192 (significant difference between the prognosis of patients with and those without BRCA1-deficient tumors (assessed either by BRCA1 promoter methylation status or by protein expression) subjected to adjuvant CT.
We changed the title from “BRCA1 promoter hypermethylation, but not BRCA1 expression, is associated with basal-like features, good prognosis, and chemosensitivity in triple-negative breast cancer” to “BRCA1 promoter hypermethylation is associated with good prognosis and chemosensitivity in triple-negative breast cancer”. As the issue of the magnitude of effect of BRCA1 promoter methylation and BRCA1 protein expression on prognosis under adjuvant chemotherapy is complex and explained in details in the manuscript, we propose to delete this part of the title in order to avoid a possible misunderstanding.
lines: 122-125: Inclusion of representative images for (retained, absent and equivocal) BRCA1 staining would be helpful to understand the scoring system used by the authors
Thank you for this suggestion. We added a supplementary figure (Supplementary Figure 1) showing the different IHC BRCA1 staining according to your comment.
lines 128-129: What selection criteria did the authors follow to choose those 153 out of 248 tumours for quantitative methylation analysis?
The comparison between methyl-specific PCR and quantitative methylation analysis was designed at the beginning of the project, in order to compare the two methods of BRCA1 promoter methylation status. At this stage of development of the project, our database was including only 153 tumors with a known BRCA1 promoter methylation status using methyl-specific PCR. Thus these cases were selected for this comparison. In the meanwhile, another 95 tumors were evaluated using the Methyl-specific PCR, raising the total number of cases to 248. No specific selection criterium was used, outside, at the time of the analysis, a known BRCA1 promoter methylation status using methyl-specific PCR. We added this information in the patients and tumors samples section of the methods.
lines 392-393: The authors clearly state that 'samples were scored as methylated only when a distinct band was present on the gel'. However it is not clearly stated when samples were scored as hypermethylated.
Thank you for this comment that highlights some discrepancies in our manuscript. The term methylation was sometimes used for hypermethylation. Thus, we changed all the “methylated / methylation” by “hypermethylated / hypermethylation” to be consistent all along the manuscript.
Minor comments
lines 23-24: The sentence 'Aberrant hypermethylation of BRCA1 promoter CpG islands, decreasing expression of BRCA1' can be detected...is not very clear. Can it be rephrased, please?
We propose to rephrase as follows: “Aberrant hypermethylation of BRCA1 promoter CpG islands induces decreased expression of BRCA1 protein. It can be detected in sporadic breast cancer without BRCA1 mutations, particularly in triple-negative breast cancers (TNBC).”
lines 347-351: Do these incubations take place at room temperature?
Thank you for this comment. We added the information in the sentence as follows: “Slides were then incubated at room temperature with antibodies raised against human BRCA1”
Reviewer 2 Report
Comments to the Author:
The authors describe BRCA1 promoter hypermethylation is associated with prognosis and chemosensitivity in triple-negative breast cancer. They find that the patients who have BRCA1 promoter hypermethylation had benefit from chemotherapy. Interestingly, they research PD-L1 expression of the patients.
The paper poses some questions.
COMMENTS
1. The authors focused on only BRCA1, not but BRCA2. If the authors could mention about that, please tell to readers a short story.
2. Line92
The authors evaluated the TNBC patients without familial BC history or known germline BRCA1 mutation. However, the rate of BRCA1/2 pathogenic mutation was unclear. The authors should mention about the data precisely.
3. Table1
Adjuvant chemotherapy regimen was unclear. How many patients received a platinum-based regimen? Platinum Salts seemed important drugs for the patients.
4. Line127-133
The authors evaluated MassARRAY testing for 153 patients. If the authors have all patients’ data of MassARRAY, the data is more robust.
5. Line178-182
The results of multivariate analysis were different from Table4. The authors should describe the results along with Table4A-4C more precisely.
Author Response
Comments to the Author:
The authors describe BRCA1 promoter hypermethylation is associated with prognosis and chemosensitivity in triple-negative breast cancer. They find that the patients who have BRCA1 promoter hypermethylation had benefit from chemotherapy. Interestingly, they research PD-L1 expression of the patients.
The paper poses some questions.
COMMENTS
1. The authors focused on only BRCA1, not but BRCA2. If the authors could mention about that, please tell to readers a short story.
Thank you for this comment. While a growing corpus of articles are reporting the frequency and biological implications of BRCA1 promoter hypermethylation, In contrast, no BRCA2 promoter methylation is implicated in breast carcinogenesis and rarely so in ovarian cancer (Collins et al., 1997; Hilton et al., 2018). Instead, BRCA2 expression has been proposed to be down-regulated by EMSY, a
chromatin remodeling protein, shown to interact with the BRCA2 transactivation domain and to repress its transcription. The EMSY gene is amplified and overexpressed in 13% BC and 17% HGSOC and this is considered as a manifestation of BRCA2 inactivation in these tumors (Hughes-Davies et al., 2003). We added a short sentence in the introduction to mention that point.
- Line92
The authors evaluated the TNBC patients without familial BC history or known germline BRCA1 mutation. However, the rate of BRCA1/2 pathogenic mutation was unclear. The authors should mention about the data precisely.
As a known BRCA1/2 pathogenic mutation was an exclusion criterion, this information was not entered in the database. Thus, no known BRCA1/2 pathogenic mutation bearer has been selected, and the exact number of pathogenic mutations, and its rate, are unknown in this database. We precised this information in the methods section.
- Table1
Adjuvant chemotherapy regimen was unclear. How many patients received a platinum-based regimen? Platinum Salts seemed important drugs for the patients.
Indeed, BRCA-deficient tumors appears more sensitive to platinum salts than BRCA-proficient ones. In our series, only one of our patients received an adjuvant platinum salts-based chemotherapy (6 cycles of docetaxel – carboplatin, due to cardiovascular comorbidities precluding the use of anthracyclines). The tumor was not hypermethylated, and expressed BRCA1 on the IHC evaluation. As per our guidelines, the other patients received either anthracycline-based chemotherapy (4 to 6 cycles of FEC100 / EC 100 / FAC65, 33.1% of the patients) or sequential anthracyclines and taxanes-based chemotherapy (3 cycles of FEC100 / EC100, followed by 3 cycles of 3-weekly docetaxel or 9 injections of weekly paclitaxel, 62.4% of the patients). Five patients received taxane-based chemotherapy, and one patient received 3 cycles of CMF followed by 3 cycles of taxanes due to a previous history of anthracyclines treatment for a hematologic malignancy. This information has been added in the results and the discussion.
- Line127-133
The authors evaluated MassARRAY testing for 153 patients. If the authors have all patients’ data of MassARRAY, the data is more robust.
The comparison between methyl-specific PCR and quantitative methylation analysis was designed at the beginning of the project, in order to compare the two methods of BRCA1 promoter methylation status. At this stage of development of the project, our database was including only 153 tumors with a known BRCA1 promoter methylation status using methyl-specific PCR. Thus these cases were selected for this comparison. In the meanwhile, another 95 tumors were evaluated using the Methyl-specific PCR, raising the total number of cases to 248. Thus, we can only present the MassARRAY data on the detailed first 153 patients.
- Line178-182
The results of multivariate analysis were different from Table4. The authors should describe the results along with Table4A-4C more precisely.
Thank you for your vigilance. Indeed, the numbers presented in the tables were right, we corrected the text accordingly and expanded the description of Table 4 in the text of the manuscript to clarify and better describe the results.
Reviewer 3 Report
In this manuscript, the authors find that BRCA1 promoter hypermethylation is associated with TNBC sensitivity to adjuvant chemotherapy, basal-like features and PD-L1 expression using a large patient cohort. It is a well designed and written paper. I recommend to accept it after minor revision.
Only one comment:
For figure 4, maybe you can present the other way: in CT group, separate the patients into subgroups based on BCRA1 promoter hypermethylation or IHC. It's more clear to see if BRCA1 promoter hypermethylation or expression is associated with sensitivity to adjuvant chemotherapy.
Author Response
Comments and Suggestions for Authors
In this manuscript, the authors find that BRCA1 promoter hypermethylation is associated with TNBC sensitivity to adjuvant chemotherapy, basal-like features and PD-L1 expression using a large patient cohort. It is a well designed and written paper. I recommend to accept it after minor revision.
Only one comment:
For figure 4, maybe you can present the other way: in CT group, separate the patients into subgroups based on BCRA1 promoter hypermethylation or IHC. It's more clear to see if BRCA1 promoter hypermethylation or expression is associated with sensitivity to adjuvant chemotherapy.
Thank you for this suggestion. Indeed, the results appear even more convincing, and the figures clearer. We changed the Figure 4 accordingly. We changed the text accordingly.
Reviewer 4 Report
The study by Jacot et al investigates BRCA1 promoter hypermethylation and its association with TNBC. The manuscript is well written and the efforts taken by the authors is appreciated. The study excels in utilizing patient samples to present evidence of BRCA1 association with TNBC and also presents evidence of association of basal-like features and PD-L1 expression.
The correlations drawn are strong and the statistical evaluations performed confirms the observed trends. The conclusions drawn from the study will help future investigators and studies to utilize PD-L1 expression, BRCA1 hypermethylation and IHC as biomarkers in TNBC.
Author Response
Comments and Suggestions for Authors
The study by Jacot et al investigates BRCA1 promoter hypermethylation and its association with TNBC. The manuscript is well written and the efforts taken by the authors is appreciated. The study excels in utilizing patient samples to present evidence of BRCA1 association with TNBC and also presents evidence of association of basal-like features and PD-L1 expression.
The correlations drawn are strong and the statistical evaluations performed confirms the observed trends. The conclusions drawn from the study will help future investigators and studies to utilize PD-L1 expression, BRCA1 hypermethylation and IHC as biomarkers in TNBC.
Thank you for your interest and your reviewing of our manuscript.
Reviewer 5 Report
The authors investigated BRCA1 methylation status and protein expression in sporadic TNBC without a known BRCA1 pathogenic variant and what their clinicopathological significance is. One of the main findings is that BRCA1 methylated tumours were sensitive to CT. Also they find that IHC is not a good predictor for methylation status. A breakdown of methylation using a different threshold of 30 or 35% is required. The authors somewhat overstate the prediction of BRCA1 hypermethylation given that it was not significant in unadjusted analysis. This needs toning down. True evidence of use of hypermethylation would of course mean evidence of PARPi sensitivity rather than just improved sensitivity to CT. This should also be recognised.
Minor comments:
- Please provide a reference for 15% of breast cancer being triple negative
- ‘Over 80% of BCs due to hereditary BRCA1 mutations are 53 TNBC [7].’ This is quite an old reference and based on a relatively small number of cases. Our data show only 26/359 (72%) BRCA1 related cancers are TNBC. The authors must provide more references to support the >80% figure or modify it to around 70-80%.
- Line 30: do you mean: BRCA1 hypermethylation associated with a reduction of IHC BRCA1 protein expression?
- Check that your acronyms are written in full for the first time (eg. Line 113: TMA; 170: OS, RFS; 194: TTR).
- Line 68 ‘Thus, BRCA1 promoter hypermethylation could be a marker of BRCA1 deficiency in the absence of BRCA1 mutation, as these two events appear mutually exclusive [20,27-31].’ This is incorrect. Promotor methylation is associated with a 5’ UTR variant at -107 in BRCA!. https://www.ncbi.nlm.nih.gov/pubmed/30075112
- Line 70 ‘Outside of BRCA1/2 deleterious mutations, there is to date no validated screening test to identify patients who may derive the most benefit from PARPi.’ Myriad market an HRD test for this purpose. If you feel this is unvalidated you need to justify the statement
- Line 78 ‘The correlation appears weak when considering protein expression (measured using Western Blot) [35] or mRNA levels [36] are considered gold standards.’ This sentence does not make sense
- Lines 151-155 and 158-161 contain very similar information. Please check.
- Line 176: is there really a trend for BRCA1 promoter hypermethylation and longer RFS (P=0.112)?
- Line 178 ‘In multivariate analysis (Table 4), shorter RFS was significantly associated with high pN stage, no adjuvant CT’ Please present results of BRCA1 IHC
- Title table 1: The table also contains tumor characteristics.
- Change mutations to pathogenic variants
- Out of interest, is there a correlation with higher methylation levels and a stronger reduction in protein expression? For instance using a cut of 30 or35% methylation. Complete promotor methylation would be expected to result in a 50% methylated signal on one copy. It is possible that much of the lack of correlation is that methylation levels of 10-30% still allow production of a sufficient amount of BRCA1 protein to give an IHC signal even if the second copy is aberrant.
- Can the authors explain why over 25% of women with TNBC did not get chemotherapy?
Author Response
Comments and Suggestions for Authors
The authors investigated BRCA1 methylation status and protein expression in sporadic TNBC without a known BRCA1 pathogenic variant and what their clinicopathological significance is. One of the main findings is that BRCA1 methylated tumours were sensitive to CT. Also they find that IHC is not a good predictor for methylation status. A breakdown of methylation using a different threshold of 30 or 35% is required. The authors somewhat overstate the prediction of BRCA1 hypermethylation given that it was not significant in unadjusted analysis. This needs toning down. True evidence of use of hypermethylation would of course mean evidence of PARPi sensitivity rather than just improved sensitivity to CT. This should also be recognised.
Thank you for these comments, which have been detailed in your 1 to 14 points below. Our point-by-point answers are presented after the associated comment. In addition, we added a sentence at the end of the abstract, in the discussion and in the conclusion to highlight the importance of evaluating the predictive value of BRCA1 promoter hypermethylation in the context of treatment by PARP inhibitors.
Minor comments:
1. Please provide a reference for 15% of breast cancer being triple negative
We inserted the 2 following references in order to justify the 15% rate for TNBCs:
- Dent, R.; Trudeau, M.; Pritchard, K.I.; Hanna, W.M.; Kahn, H.K.; Sawka, C.A.; Lickley, L.A.; Rawlinson, E.; Sun, P.; Narod, S.A. Triple-negative breast cancer: clinical features and patterns of recurrence. Clinical cancer research : an official journal of the American Association for Cancer Research 2007, 13, 4429-4434, doi:10.1158/1078-0432.CCR-06-3045.
- Elias, A.D. Triple-negative breast cancer: a short review. Am J Clin Oncol 2010, 33, 637-645, doi:10.1097/COC.0b013e3181b8afcf.
2. ‘Over 80% of BCs due to hereditary BRCA1 mutations are 53 TNBC [7].’ This is quite an old reference and based on a relatively small number of cases. Our data show only 26/359 (72%) BRCA1 related cancers are TNBC. The authors must provide more references to support the >80% figure or modify it to around 70-80%.
Thank you for this comment. Indeed, the “over 80%” was quite high. We added the following, more recent, references, and modified the text to “70 to 80%”.
Patterns of recurrence and metastasis in BRCA1/BRCA2-associated breast cancers. Song Y, Barry WT, Seah DS, Tung NM, Garber JE, Lin NU.Cancer. 2020 Jan 15;126(2):271-280. doi:10.1002/cncr.32540. Epub 2019 Oct 3.
Genetic susceptibility to triple-negative breast cancer. Stevens KN, Vachon CM, Couch FJ.Cancer Res. 2013 Apr 1;73(7):2025-30. doi: 10.1158/0008-5472.CAN-12-1699. Epub 2013 Mar 27.
3. Line 30: do you mean: BRCA1 hypermethylation associated with a reduction of IHC BRCA1 protein expression?
Indeed, the sentence was not sufficiently precise, as it was not indicating the sense of the association. We added the “reduction” information in the introduction.
4. Check that your acronyms are written in full for the first time (eg. Line 113: TMA; 170: OS, RFS; 194: TTR).
We carefully checked the acronyms in the text and corrected as requested.
5. Line 68 ‘Thus, BRCA1 promoter hypermethylation could be a marker of BRCA1 deficiency in the absence of BRCA1 mutation, as these two events appear mutually exclusive [20,27-31].’ This is incorrect. Promotor methylation is associated with a 5’ UTR variant at -107 in BRCA!. https://www.ncbi.nlm.nih.gov/pubmed/30075112
Thank you for this interesting information. We mitigated the sentence as follows:
« …as these two events appear almost mutually exclusive [24,31-35], outside of the recently described association between a dominantly inherited 5' UTR variant, classified as likely pathogenic, and BRCA1 promoter hypermethylation [36].”
6. Line 70 ‘Outside of BRCA1/2 deleterious mutations, there is to date no validated screening test to identify patients who may derive the most benefit from PARPi.’ Myriad market an HRD test for this purpose. If you feel this is unvalidated you need to justify the statement
Thank you for this comment. We corrected the sentence to specify that our comment is only addressing the question of PARPi prediction in breast cancer patients. Indeed, Myriad myChoice CDx achieved a level of evidence, but, to date, only in ovarian cancer patients. To date, the only predictive biomarkers of PARPi efficacy in breast cancer, based on the OLYMPIA and EMBRACA studies, are BRCA1, BRCA2 and PALB2 mutations.
7. Line 78 ‘The correlation appears weak when considering protein expression (measured using Western Blot) [35] or mRNA levels [36] are considered gold standards.’ This sentence does not make sense
We clarified the sentence as follows : The correlation between levels of BRCA1 transcription and Western Blot protein quantification has been reported as weak [42]. In the study by Al-Mula et al., the same weak correlation was found between IHC protein evaluation and mRNA levels quantification using real-time RT-PCR [43].
8. Lines 151-155 and 158-161 contain very similar information. Please check.
In order to simplify this section, with, as noted, close content, we modified the text as follows:
The Venn diagram (Figure 3) summarizes the cross-distribution of the evaluated population across the three proposed biomarkers of BRCA1 deficiency, i.e. BRCA1 protein expression, BRCA1 promoter hypermethylation, and basal-like phenotype. Among the 248 TNBC cases, three tumors with missing IHC basal marker statuses were excluded from the analysis and only 14.7% (36/245) of samples did not exhibit any of the three BRCA1 deficiency markers. There is an overlap of TNBC BRCA1-deficient specimens assessed by the three different measures; however, only 25 (10.2%) of the tumors displayed the 3 biomarkers of BRCA1 deficiency.
9. Line 176: is there really a trend for BRCA1 promoter hypermethylation and longer RFS (P=0.112)?
Indeed, this p value is not statistically significant; however the sentence was designed to “pinpoint” a non-significant difference in the overall population and to highlight the presence of a statistical difference in the chemotherapy group. In order to avoid any risk of confusion or overstating, we modified the sentences as follows:
Results of the univariate analysis (regarding RFS and OS) in the overall population are summarized in Table 3. High pT and pN stages, a lobular subtype, and absence of adjuvant CT were significantly associated with shorter RFS. Five years RFS rate was 79.9 and 71% (p=0.112) in BRCA1 promoter hypermethylated and non-hypermethylated groups, respectively. For OS, age, pT and pN stages, and use of adjuvant CT were significant determinants.
10. Line 178 ‘In multivariate analysis (Table 4), shorter RFS was significantly associated with high pN stage, no adjuvant CT’ Please present results of BRCA1 IHC
For the multivariate analysis, we selected variables of interest as variables with p-values <0.15 in univariate analysis (we added this information in the statistical analysis section of the methods). Thus, BRCA1 protein expression was not considered in the multivariate analysis presented in figure 4. In order to answer the question, we performed a second multivariate analysis integrating this variable, in order to test the robustness of the results (Supplementary Table 1). BRCA1 protein expression by IHC was not a statistically significant determinant of prognosis, whatever the population considered, and the previously identified variables remained significant.
11. Title table 1: The table also contains tumor characteristics.
Thank you. We modified the title accordingly (Patients and tumors characteristics)
12. Change mutations to pathogenic variants
We modified this information all over the manuscript.
13. Out of interest, is there a correlation with higher methylation levels and a stronger reduction in protein expression? For instance using a cut of 30 or35% methylation. Complete promotor methylation would be expected to result in a 50% methylated signal on one copy. It is possible that much of the lack of correlation is that methylation levels of 10-30% still allow production of a sufficient amount of BRCA1 protein to give an IHC signal even if the second copy is aberrant.
Thank you for this really interesting and useful question. Indeed, based on this idea, we checked our series for tumors with high percentages of methylation. This analysis faced 2 difficulties.
First, the comparison between methyl-specific PCR and quantitative methylation analysis was designed at the beginning of the project, in order to compare the two methods of BRCA1 promoter methylation status. At this stage of development of the project, our database was including only 153 tumors with a known BRCA1 promoter methylation status using methyl-specific PCR. Thus, the use of a quantitative cut-off for methylation will significantly reduce the statistical power of the evaluation.
Second, as shown in Figure 2, tumors with high percentages of methylation are rare events in our series: 12 tumors (7.84%) for the 30% cut-off, 9 (5.88%) for the 35% cut-off, and 2 (1.31%) for the 50% cut-off.
Such low frequency precludes all robust statistical evaluation, outside by increasing dramatically the number of samples.
We added this information in the discussion as one of the limits of the present work, and as a suggestion for future works.
14. Can the authors explain why over 25% of women with TNBC did not get chemotherapy?
As noted L328, “Each individual treatment proposal was in accordance with our institution guidelines”, and must be put into context of adjuvant medical decisions in the considered period (202 – 2010), as, in the early 2000’s, guidelines regarding TNBC adjuvant treatments were somewhat less oriented to a generalization of adjuvant chemotherapy indications. The multidisciplinary tumor board discussed every specific patient, and most of the reasons for not considering adjuvant chemotherapy were age, T and N status, and pathological grade. In addition, some patients refused an indicated adjuvant treatment. Indeed, if evaluating the main differences between the population that received or not an adjuvant chemotherapy in our series, greater age (p<0.001) and low pathological grade (p=0.002) were significantly associated with the absence of adjuvant chemotherapy. However, as these variables were tested in the multivariate analysis, a confounding effect can be ruled out. We added a sentence in the discussion regarding this point.
Round 2
Reviewer 5 Report
The authors have responded well to the review